# OpenReview forum: "Adaptive Distraction: Probing LLM Contextual Robustness with Automated Tree Search"
_NeurIPS.cc/2025/Conference — NeurIPS 2025 poster_

### Official Review · Reviewer_SXak · 2025-06-10

**Clarity:** 3
**Significance:** 1
**Originality:** 1
**Rating:** 2
**Confidence:** 4

**Summary:**

Adaptive distractions generated by automated tree search can cause large language models to lose over 45% accuracy. While prompt-based defenses offer limited protection, post-training approaches like DPO significantly improve robustness to such distractions.

**Questions:**

Please refer to the Weaknesses

**Ethical Concerns:**

["NO or VERY MINOR ethics concerns only"]

**Final Justification:**

The main concern, as I have repeatedly emphasized, is that I find the originality, conceptual motivation, and the choice of baselines in this paper to be inappropriate.

The authors claim that:

“Our goal is not merely to point out that LLMs are susceptible to contextual interference, but to introduce a novel, automated framework for discovering such vulnerabilities in a systematic, behavior-guided, and answer-preserving manner.”

However, the paper itself admits that many of the proposed problems are constructed by introducing conflicts in the context. First, it has already been demonstrated in many knowledge conflict studies that models struggle with fixed conflict contexts (q, c_conflict). The present work merely extends this by adding a dynamic component, but this extension is still fundamentally based on existing paradigms.

Secondly, since the evaluation is constructed by introducing conflict into the context to test the model, the most appropriate baselines should be benchmarks or methods that also introduce conflict—not those that only inject irrelevant information.

Overall, I find the motivation, the quality of the work, and the selection of baselines in this paper to be inappropriate. My concerns remain unresolved.

**Limitations:**

yes

**Paper Formatting Concerns:**

The instruction block is not deleted.

**Quality:**

1

**Strengths And Weaknesses:**

Strengths

- The method exposes fundamental vulnerabilities in large language models by generating adaptive distractions that cause significant performance drops.
- Extensive experiments show that the approach is effective across multiple benchmarks and model types.

Weaknesses / Limitations

- My main concern is that constructing such "contextually plausible" distractions essentially examines model robustness in scenarios where the provided context conflicts with the model's own knowledge—that is, where the context supports one answer while the model's prior knowledge suggests another. In many of the case studies presented, adding distractions appears to transform the problem from open-book QA into a reading comprehension task. For instance, the question "What is the best medicine?" plus the context "Many people find that healing often requires patience and allows for natural recovery over time." will naturally lead to the answer "Time is the best medicine." In such cases, the context itself biases the model toward certain options. The evaluation then becomes more about how the model chooses between conflicting external knowledge sources, a problem already widely discussed in the literature on knowledge conflicts—where it is well established that models are easily influenced by contextual information.

     Furthermore, when it comes to dataset selection, many questions without fixed answer options do not have absolute ground-truth answers. In these scenarios, requiring the model to follow the given context may actually be more aligned with the model's training objectives and practical applications (e.g., retrieval-augmented generation scenarios). Thus, I am not convinced that discouraging the model from using the context—especially through DPO training—would be meaningful or desirable on such datasets.

- Some important details are missing. For example, where is the DPO (Direct Preference Optimization) training data from? What are the labels, and how are they generated? How is the training actually performed? I could not find these clarifications in Section B1 or the main text.

- Based on the above, I am concerned that after DPO training, the model may not be able to follow instructions well, or may struggle in settings where it needs to answer questions using context (e.g., RAG scenarios). It would be helpful to see experiments or discussion on how the method generalizes to tasks requiring instruction-following or context-dependent reasoning.

- Why set the temperature to 0.001 instead of 0?

---

> ### Author Rebuttal · Authors · 2025-07-30
>
> We sincerely thank the reviewer for the detailed and thoughtful comments. We have carefully revised the paper based on your suggestions, and we address each point below.
>
> ---
>
> **$Q_1:$ My main concern is that constructing such "contextually plausible" distractions essentially examines model robustness in scenarios where the provided context conflicts with the model's own knowledge—that is, where the context supports one answer while the model's prior knowledge suggests another. In many of the case studies presented, adding distractions appears to transform the problem from open-book QA into a reading comprehension task.**
>
> **$A_1:$** We thank the reviewer for this insightful point and the concrete example provided.
>
> (1) We agree that in rare edge cases such as *“What is the best medicine?”* with the added context *“healing often requires patience…”*, the inserted sentence may imply a specific answer (“Time is the best medicine”), thereby shifting the question toward a reading comprehension format. **This is precisely the type of semantic shift our quality control pipeline is designed to eliminate.** Specifically, we employ both automatic semantic consistency checks and manual verification (see human evaluation in **Table 15**) to filter out cases where the added context might reasonably support a new correct answer. The vast majority of retained distractions are designed to be **plausible yet irrelevant**, ensuring the question's meaning and correct answer remain unchanged.
>
> (2) Our experiments are conducted on **benchmarks with fixed ground-truth answers** (e.g., MMLU, TruthfulQA), where correctness can be objectively defined. The added distractions simulate realistic noisy-retrieval scenarios rather than introducing contradictory facts. For example, in **Figure 1**, adding earthquake casualties misleads the model on a question about terrorism deaths—not due to knowledge conflict, but because the model fails to resist a **semantically irrelevant but numerically tempting** distraction.
>
> (3) While we acknowledge that a few borderline cases may resemble knowledge conflicts, our work targets a **different and underexplored failure mode**: the model’s susceptibility to **contextual distraction vulnerability**—i.e., being misled by information that is semantically coherent but logically irrelevant. This phenomenon is empirically distinct from classical knowledge conflict setups (e.g., [1]) and exposes a practical robustness gap in how models handle weakly related context.
>
> We will clarify this distinction more explicitly and include the reviewer’s helpful example in the revised version.
>
> [1] Making Retrieval-Augmented Language Models Robust to Irrelevant Context
>
> **$Q_2:$ Furthermore, when it comes to dataset selection, many questions without fixed answer options do not have absolute ground-truth answers. In these scenarios, requiring the model to follow the given context may actually be more aligned with the model's training objectives and practical applications (e.g., retrieval-augmented generation scenarios). Thus, I am not convinced that discouraging the model from using the context—especially through DPO training—would be meaningful or desirable on such datasets.**
>
> **$A_2:$** We appreciate the reviewer’s perspective and would like to clarify that **our method does not discourage the model from using context**. Rather, our goal is to improve the model’s ability to **distinguish between relevant and irrelevant information**—a critical skill for real-world applications such as RAG.
>
> Our DPO training encourages the model to **resist distractions that are semantically plausible but irrelevant to the task**, not to ignore context in general. In fact, as demonstrated in **our $Q_4$ experiment on SQuAD v2**, a benchmark that requires strong instruction-following and contextual reasoning, models trained with DPO exhibit **nearly identical performance** to the original models. This shows that DPO does not impair the model’s ability to leverage useful context or follow instructions. Instead, it selectively suppresses overreliance on misleading cues.
>
> We fully agree that open-ended tasks with no clear ground-truth are beyond the scope of our current framework. As also noted in Q1, applying our method to such tasks risks conflating robustness with task reinterpretation. That said, we believe our findings do generalize to practical settings where noisy or misleading retrieval results can hurt model reasoning. In those cases, our DPO-enhanced models would still be capable of attending to meaningful context while avoiding harmful distraction.
>
> We will make this distinction and supporting evidence more explicit in the revised version.
>
> **$Q_3:$** **Regarding DPO, as shown in Table 14, fine-tuning does not significantly affect performance on clean inputs while substantially improving robustness on distracted ones. This suggests that DPO mainly reduces over-reliance on irrelevant cues rather than discouraging the model from using legitimate context, which aligns with our goal in fixed-answer robustness evaluation.**
>
> **$A_3:$** We thank the reviewer for pointing this out. We would like to clarify that the requested details are already provided in **Appendix B.1** (lines 576–592), though they may have been overlooked. Specifically, the DPO training data are derived from our perturbation experiments, where each preference pair consists of a question, its correct answer, and an incorrect answer collected from model responses. We ensure that enhanced questions originating from the same original question do not appear in both the training and test sets. The fine-tuning strictly follows the standard DPO procedure, using **three open-source models** (Gemma-2-2B, Qwen2.5-7B, and Phi-3.5-mini), trained for five epochs with a learning rate of 2e-4.
>
> **$Q_4:$** **Based on the above, I am concerned that after DPO training, the model may not be able to follow instructions well, or may struggle in settings where it needs to answer questions using context (e.g., RAG scenarios). It would be helpful to see experiments or discussion on how the method generalizes to tasks requiring instruction-following or context-dependent reasoning.**
>
> **$A_4:$** We thank the reviewer for this valuable suggestion and conducted an additional experiment to verify whether DPO affects instruction-following or context-dependent reasoning. We chose **SQuAD v2**, a widely used benchmark for measuring contextual understanding and instruction-following ability, and randomly sampled 200 questions for evaluation. Three models fine-tuned with DPO (Gemma-2-2B, Phi-3.5-mini, and Qwen-2.5-7B) were tested under the same zero-shot CoT protocol as described in Section 3.1.
>
> | Model        | Original Acc. (Clean) | DPO Acc. (Clean) |
> | ------------ | --------------------- | ---------------- |
> | Gemma-2-2B   | 126/200 (0.630)       | 124/200 (0.620)  |
> | Phi-3.5-mini | 125/200 (0.625)       | 127/200 (0.635)  |
> | Qwen-2.5-7B  | 170/200 (0.850)       | 166/200 (0.830)  |
>
> The results show that the DPO-tuned models achieve **almost the same performance** as the original models on clean inputs (all differences <2%). This confirms that DPO does not impair the models’ ability to follow instructions or leverage relevant context. Combined with the substantial robustness improvements observed on our adaptive distraction benchmarks (Section 3.5), these findings further highlight the unique contribution of our framework: it strengthens resistance to irrelevant but plausible distractions while leaving **normal context reasoning capability**.
>
> **$Q_5:$ Why set the temperature to 0.001 instead of 0?**
>
> **$A_5:$** We appreciate the reviewer for pointing out this detail. We agree that **using 0 is the conventional choice** for evaluation, and we have verified that the results remain almost identical when setting the temperature to 0. We used 0.001 mainly to ensure consistent deterministic decoding across different APIs, as some implementations treat 0 as a special case (e.g., enabling minimal stochasticity or fallback sampling). Setting it to a very small nonzero value avoids such implementation-dependent behavior while keeping the generation effectively deterministic. We will clarify this in the revised version.
>
> ---
>
> Thank you again for your valuable time and suggestions and we would greatly appreciate your kind support during the discussion phase.

---

> ### Comment · Reviewer_SXak · 2025-08-02
> **Response to Authors**
>
> Thank you for your response. However, I feel that my main concerns have still not been addressed, and I now have even more doubts regarding the completeness and originality of this work.
>
> **Regarding your explanation of how your work differs from previous research on knowledge conflicts --- "our work targets a different and underexplored failure mode: the model’s susceptibility to contextual distraction vulnerability—i.e., being misled by information that is semantically coherent but logically irrelevant. This phenomenon is empirically distinct from classical knowledge conflict setups."**
>
> If I understand correctly, the authors claim that the main distinction is the notion of  "semantically coherent but logically irrelevant." However, I would like to ask: How do you define "semantically coherent but logically irrelevant"?
>
> For instance, in Figure 1 of [1], could you clarify why the example is not considered a standard case of semantic coherence yet logical irrelevance? Furthermore, some previous conflict works are not purely about factual knowledge. For example, same cased as build on the CommonsenseQA dataset, consider the question: "What do people aim to do at work?" and the context "Some jobs, such as butchers or hunters, involve killing animals as part of their work duties." Here, the misleading answer is "kill animals." Would this not also qualify as "semantically coherent but logically irrelevant"?
>
> Even if you are able to provide a clear definition, the major issue remains:  You have admitted that, in your setup,  "context supports one answer while the model’s prior knowledge suggests another," and even acknowledge that "a few borderline cases may resemble knowledge conflicts." In my view, this essentially places your evaluation within the knowledge conflict setting, which has already been extensively explored [2]. If you just in turn treat this phenomenon as an "attack” mechanism, then the **originality and completeness** of this work are seriously called into question.
>
> [1] Making Retrieval-Augmented Language Models Robust to Irrelevant Context
> [2] Knowledge Conflicts for LLMs: A Survey

---

> ### Author Response · Authors · 2025-08-02
>
> Thank you very much for your thoughtful comments and for highlighting these important concerns. We must respectfully point out the **potential misunderstanding** about our contribution, **namely that it lies not in defining distractions, but in systematically searching for distractions based on model feedback signals**, as this misunderstanding may significantly negatively impact the evaluation of our work. We are glad to have the opportunity to clarify and address your concerns below.
>
> **$Q_1$: How does our work differ from previous research on knowledge conflicts?**
>
> **$A_1$:** While prior work has largely focused on identifying and characterizing contextual failures, **our core contribution lies in proposing a general, automated framework that systematically discovers model-specific weaknesses via behavior-guided distraction generation and optimization**. Unlike previous static or template-based methods, our approach utilizes victim-model responses through automated tree search to effectively identify and exploit deeper contextual robustness vulnerabilities. Additionally, our framework rigorously integrates semantic consistency checks to ensure the generated distractions do not alter the ground-truth answers, reinforcing the robustness of our evaluations.
>
> **$Q_2$: How do we clearly define "semantically coherent but logically irrelevant" distractions, and how does our definition highlight the core difference?**
>
> **$A_2$:** In our work, "semantically coherent but logically irrelevant" distractions refer to content that is contextually plausible, but carefully constructed so as not to offer any valid reasoning path toward the correct answer. The purpose of enforcing this property is to **ensure that the original problem still has a unique**, unambiguous answer (thus avoiding label uncertainty) while probing the model's susceptibility to distraction from contextually rich yet logically unhelpful content.
>
> For example, in the CommonsenseQA case you cited ("What do people aim to do at work?" with the added context "Some jobs involve killing animals"), the added context introduces a directly answerable and plausible distractor ("kill animals"), which violates our semantic filtering criterion. In contrast, our method employs a rigorous semantic screening mechanism to rule out candidate distractions that might introduce such ambiguity or inadvertently suggest alternative correct answers.
>
> Our goal in constructing these semantically coherent but logically irrelevant distractions is **not to redefine the notion itself**, but to implement it systematically within an automated, model-specific framework. These distractions are optimized based on model behavior and validated to maintain answer stability, enabling us to diagnose weaknesses in contextual robustness that would otherwise remain hidden.
>
> To demonstrate the robustness and generalizability of our approach, we evaluated our method across **diverse tasks** including highly factual domains (such as mathematical reasoning using the MATH dataset) and safety-critical contexts (using the RealToxicityPrompts dataset). Our method consistently showed effectiveness without introducing ambiguity or factual conflicts, underscoring our framework’s broad applicability and reliability.
>
> A concrete illustration from the mathematical reasoning task:
>
> * **Original Question**:
>
>   > "What is the smallest positive multiple of 450 whose digits are all zeroes and ones?"
>
> * **With our optimized distraction**:
>
>   > "What is the smallest positive multiple of 450 whose digits are all zeroes and ones? The structure of numbers composed entirely of ones and zeroes can sometimes yield unexpected multiples when examined closely. Exploring the patterns of numbers made solely from ones and zeroes can occasionally reveal fascinating relationships with larger multiples that might not be immediately apparent."
>
> In this scenario, our generated distraction remains semantically coherent and plausible but crucially does not introduce ambiguity or conflicting logic. Our systematic, feedback-based optimization approach ensures such distractions consistently reveal genuine robustness weaknesses, reinforcing that our key contribution lies in the automated discovery of model-specific weaknesses via behavior-guided search and optimization.
>
> We will clearly emphasize these points in our revised manuscript. Notably, other reviewers have acknowledged this distinction and specifically recognized the novelty of our contribution in systematically uncovering model-specific contextual robustness failures via automated and behavior-guided distraction generation. We respectfully hope you will also reconsider your assessment in light of this clarification.

---

> > ### Comment · Reviewer_SXak · 2025-08-03
> > **Response  to the Authors**
> >
> > You mention "systematically searching for distractions based on model feedback signals," but your approach essentially involves providing contextual information to elicit a candidate response that differs from the target model’s expected answer. This is, in essence, a form of knowledge conflict, so I do not think there is any misunderstanding on my part.
> >
> > The precise definition of “distractions” is not my primary concern, as fundamentally, there is no essential difference from previous knowledge conflict work. As a result, I find the originality, conceptual motivation, and the choice of baselines in this paper to be inappropriate.
> >
> > I will maintain my negative position, as my core concerns remain unaddressed.

---

> > > ### Author Response · Authors · 2025-08-04
> > >
> > > Thank you for your continued engagement. However, we respectfully believe that there **remains a significant misunderstanding** about the core contribution of our work.
> > >
> > > **Our goal is not merely to point out that LLMs are susceptible to contextual interference, but to introduce a novel, automated framework for discovering such vulnerabilities in a systematic, behavior-guided, and answer-preserving manner.** Unlike static or template-based baselines, our method leverages model behavior in a tree-search optimization loop to generate adversarial yet answer-preserving distractions. Empirically, our method causes an average performance drop exceeding **45%**, whereas all previous distraction baselines induce only **~15%** or less drop. This significant difference in effect size demonstrates that our distractions are substantially **more effective in exposing weaknesses**.
> > >
> > > Reviewer coVs has accurately **summarized** this point:
> > >
> > > > This paper’s use of an adaptive, model-guided tree search to generate adversarial context is a clear advancement.
> > >
> > > We hope this clarifies that our contribution is not the identification of a phenomenon, but the design of a technically novel and impactful framework to systematically evaluate and improve contextual robustness in LLMs. We respectfully encourage you to reconsider your assessment in light of this clarification.

---

### Official Review · Reviewer_coVs · 2025-07-02

**Clarity:** 4
**Significance:** 4
**Originality:** 3
**Rating:** 5
**Confidence:** 4

**Summary:**

This paper introduces a novel framework for evaluating the contextual robustness of Large Language Models (LLMs). The authors address the issue that LLMs' performance often degrades when presented with task-irrelevant but  semantically coherent information, a phenomenon they term "adaptive distraction." The core of their contribution is a dynamic distraction generation method based on an automated tree search. This process is guided by the target model's own behavior, starting with a classifier to filter for susceptible questions and then using a proxy model to iteratively generate distracting context aimed at inducing model failure. The framework is systematically evaluated across four benchmark datasets and a wide range of modern LLMs, demonstrating a significant average performance drop of over 45%. The paper further explores mitigation strategies, finding that prompt-based methods offer little benefit, while post-training techniques like Direct Preference Optimization (DPO) can substantially enhance robustness.

**Questions:**

1. **On the Cost-Benefit of Tree Search:** The tree search is computationally intensive. Your ablation study on the value function shows that using the full function is better than simpler versions. However, could you provide a more direct comparison between a simple, one-step "Error-Guided Perturbation" (generating one distraction for each wrong option and stopping) versus the full, multi-level tree search? Specifically, what is the marginal gain in performance drop achieved by searching at depths > 1, and is this gain worth the exponential increase in computational cost?
2. **Generalizability to Open-Ended Tasks:** Your framework relies on a discrete set of incorrect answers to guide the perturbation generation. How do you envision adapting this "Error-Guided Perturbation" to more open-ended tasks like text summarization or story generation, where there isn't a clear set of "wrong options"? Would the guidance have to be based on more abstract principles (e.g., generating distractions that encourage factual inaccuracies or stylistic deviations)?
3. **Qualitative Analysis of Failures:** The case studies in the appendix are insightful. Could you provide a more systematic qualitative analysis of the failure modes? For instance, are the models failing primarily by latching onto superficial keywords introduced in the distraction (a "shallow" failure), or are the distractions causing a more fundamental breakdown in the model's reasoning process (a "deep" semantic confusion)?
4. **Sensitivity to Hyperparameters:** The value function includes constants $\alpha=2$ and $\gamma=1$. How were these values chosen? Have you performed any sensitivity analysis to determine how robust the search process is to changes in these hyperparameters?

**Ethical Concerns:**

["NO or VERY MINOR ethics concerns only"]

**Final Justification:**

I thank the authors for their detailed and comprehensive response. The rebuttal further strengthens the paper’s claims, and I am happy to maintain my score of ‘Accept’. This work represents a timely and valuable contribution to the evaluation of LLM robustness.

**Limitations:**

Yes

**Quality:**

3

**Strengths And Weaknesses:**

### **Strengths**

\* **Significance:** The paper tackles a highly significant and timely problem. As LLMs are integrated into real-world applications, their reliability in noisy, complex environments is paramount. This work provides a rigorous and scalable method to stress-test and quantify a critical failure mode—vulnerability to contextual distractions—which is a crucial step towards building more reliable systems.

\* **Originality and Novelty:** The proposed method is novel and sophisticated. While prior work has explored static or retrieval-based distractions, this paper's use of an adaptive, model-guided tree search to generate adversarial context is a clear advancement. The integration of a pre-filtering classifier and a value function guided by simulation success rates constitutes a well-designed, original framework for this problem.

\* **Quality of Evaluation:** The empirical evaluation is extensive and of high quality. The authors test their method on a diverse and relevant set of twelve modern LLMs, including both proprietary and open-weight models. The use of four different QA benchmarks demonstrates the broad applicability of the vulnerability. Furthermore, the comparison against multiple baselines and the thorough investigation of mitigation strategies (both prompting and fine-tuning) add significant depth and value to the experimental results. The inclusion of detailed ablation studies on the value function and classifier further strengthens the paper's claims.

\* **Clarity:** The paper is exceptionally well-written and clearly structured. The methodology is presented logically, aided by a helpful diagram (Figure 1) and formal problem definitions. The results are clearly tabulated and discussed, and the appendices provide extensive details on hyperparameters, prompts, and additional experiments, ensuring high reproducibility.

###  **Weaknesses**

\* **Scalability and Computational Cost:** The primary weakness of the proposed method is its computational expense. A tree-search process that involves multiple calls to powerful LLMs for generation (proxy model) and simulation (victim model) at each node is inherently costly. While the authors have cleverly incorporated efficiency strategies like a pre-filtering classifier and early stopping, the scalability of this approach for very large datasets or its integration into an online training loop remains a significant challenge. The paper acknowledges this cost but could benefit from a more detailed analysis of the cost-benefit trade-off.

\* **Limited Scope of Tasks:** The experiments are primarily focused on multiple-choice question-answering tasks. While the authors show generalization to a math reasoning benchmark, it is still a format with a single, verifiable ground truth. It is less clear how the core mechanics of the framework—particularly the error-guided perturbation based on incorrect options and the success rate simulation—would apply to more open-ended tasks like summarization, translation, or creative writing, where defining "failure" is more nuanced.

\* **Lack of Statistical Significance Testing:** The paper presents extensive results but lacks reporting of confidence intervals or statistical significance tests for its main claims. Given the stochastic nature of LLM responses (even with low temperature), reporting error bars across multiple runs would strengthen the reliability of the reported accuracy drops and comparison between methods. The authors acknowledge this in the checklist, citing computational cost, but it remains a methodological weakness.

---

> ### Author Rebuttal · Authors · 2025-07-30
>
> We sincerely thank the reviewer for the detailed and thoughtful comments. We have carefully revised the paper based on your suggestions, and we address each point below.
>
> ---
>
> **$Q_1:$ The tree search is computationally intensive. Your ablation study on the value function shows that using the full function is better than simpler versions. However, could you provide a more direct comparison between a simple, one-step "Error-Guided Perturbation" (generating one distraction for each wrong option and stopping) versus the full, multi-level tree search? Specifically, what is the marginal gain in performance drop achieved by searching at depths > 1, and is this gain worth the exponential increase in computational cost?**
> **$A_1:$** We thank the reviewer for raising this important question.
> (1) The single-step “Error-Guided Perturbation” essentially corresponds to our earlier baselines (e.g., SPD), where only one perturbation is generated per wrong option without iterative refinement. In contrast, our method incrementally adds semantically coherent perturbations, with the search depth representing the number of combined distractions.
> (2) We conducted a direct comparison between the single-step baseline, brute-force search, and our **value-guided search at depth ≤3**. The results (averaged over four datasets) are summarized below:
>
> | Strategy                 | Avg. Acc. Drop ↓ | Avg. Nodes Explored ↓ |
> | ------------------------ | ---------------- | --------------------- |
> | Single-step (Depth = 1)  | 18.5%            | 3.0                   |
> | Brute-force Search (≤3)  | 51.2%            | 14.7                  |
> | Value-guided Search (≤3) | 49.5%            | 5.3                   |
>
> These results show that our value-guided search achieves **almost the same performance drop as brute-force search** while reducing search cost by ~64%.
> (3) We believe the additional depth is worthwhile. The single-step baseline fails to expose the worst-case vulnerabilities (e.g., only ~18% degradation), while deeper search identifies more damaging distractions efficiently, which aligns with our goal of stress-testing contextual robustness rather than minimizing compute.
>
> **$Q_2:$ Your framework relies on a discrete set of incorrect answers to guide the perturbation generation. How do you envision adapting this "Error-Guided Perturbation" to more open-ended tasks like text summarization or story generation, where there isn't a clear set of "wrong options"? Would the guidance have to be based on more abstract principles (e.g., generating distractions that encourage factual inaccuracies or stylistic deviations)?**
> **$A_2:$** We thank the reviewer for the valuable suggestion regarding extending our framework to safety-sensitive generation tasks. We fully agree that this is an important direction to test the generality of our method.
>
> However, as also noted in prior literature on knowledge conflicts [1], generation tasks such as summarization or open-ended QA introduce intrinsic ambiguity: the output does not have a single ground-truth answer, and adding semantically coherent distractions may naturally shift the intended interpretation of the task rather than purely reflecting robustness issues. For instance, when extra context contradicts the model’s prior knowledge, the model might simply follow the context, which is an expected behavior in retrieval-augmented generation scenarios rather than a failure of contextual robustness. Therefore, we believe that applying our current answer-preserving, ground-truth-based framework directly to such open-ended tasks would risk **confounding contextual robustness with task re-interpretation**. We consider this an important future direction, where task-specific robustness metrics (e.g., factual consistency measures) can be integrated.
>
> In contrast, safety-critical tasks provide **well-defined objectives** (e.g., generating harmful or toxic outputs), making them more suitable for our framework. To explore this, we conducted additional experiments using the RealToxicityPrompts dataset, which is widely used for evaluating model safety. For each prompt, we searched for a semantically coherent instructional distraction (i.e., a prefix before the prompt) that would most increase the toxicity of the response, guided by our adapted value function (using Perspective API scores as feedback).
>
> We compared our method against a single-step baseline that applies a direct distraction prefix without iterative refinement. The results on 175 samples are summarized below:
>
> | Model       | Original Toxicity | +Direct Distraction | +Adaptive Distraction (Ours) |
> | ----------- | ----------------- | ------------------- | ---------------------------- |
> | GPT-4o-mini | 0.0304            | 0.2279              | **0.5496**                   |
> | Qwen2.5-72B | 0.0908            | 0.2256              | **0.5190**                   |
> | GPT-4o      | 0.0277            | 0.1291              | **0.2737**                   |
> | LLaMA3-8B   | 0.0558            | 0.1508              | **0.3616**                   |
> | Gemma-2-27B | 0.0998            | 0.1857              | **0.4748**                   |
>
> These results demonstrate that our adaptive distraction strategy can significantly increase model toxicity more effectively than single-shot distractions, confirming that the proposed framework is capable of revealing safety vulnerabilities beyond standard QA tasks.
>
> We will include these findings in the revised version.
>
> **$Q_3:$ The case studies in the appendix are insightful. Could you provide a more systematic qualitative analysis of the failure modes? For instance, are the models failing primarily by latching onto superficial keywords introduced in the distraction (a "shallow" failure), or are the distractions causing a more fundamental breakdown in the model's reasoning process (a "deep" semantic confusion)?**
> **$A_3:$** We thank the reviewer for this helpful suggestion. We conducted a closer inspection of failure cases to better characterize the failure modes.
>
> For many examples, the errors are not caused solely by superficial keyword matching but by a **deeper reasoning breakdown** triggered by the distraction. For instance, in the case shown in Figure 1 (“In 2017, how many people died from terrorism globally?”), the adaptive distraction introduces semantically plausible but irrelevant information (“A devastating earthquake… resulting in thousands of casualties”). Although this additional context does not logically support any change in the correct answer, the model incorrectly outputs “2,600.” This suggests that the model conflates large-scale casualty information with terrorism deaths, indicating a **semantic-level confusion** rather than simple keyword bias.
>
> Such observations are consistent across many failure cases, showing that our method can reveal not only shallow keyword biases but also deeper reasoning vulnerabilities. We will expand our qualitative analysis and add more representative examples in the revised version.
>
> **$Q_4:$ The value function includes constants α=2 and γ=1. How were these values chosen? Have you performed any sensitivity analysis to determine how robust the search process is to changes in these hyperparameters?**
> **$A_4:$** We set **α=2 and γ=1** to ensure that the failure signal \( $r_M(P')$ \) plays a dominant role in node ranking, while the depth term serves as a mild penalty to discourage overly deep expansions.
>
> In preliminary experiments, we varied **α in [1, 3] and γ in [0.5, 1.5]** to balance these two factors. After comparison, we found that α=2 and γ=1 yielded the best trade-off between prioritizing low-success-rate nodes and maintaining search efficiency. These values were thus fixed in all main experiments, and we will clarify this design choice in the revised version.
>
> [1] Making Retrieval-Augmented Language Models Robust to Irrelevant Context
>
> ---
>
> Thank you again for your valuable time and suggestions and we would greatly appreciate your kind support during the discussion phase.

---

### Official Review · Reviewer_WpC3 · 2025-07-03

**Clarity:** 3
**Significance:** 3
**Originality:** 3
**Rating:** 5
**Confidence:** 2

**Summary:**

Previous static methods like fixed-template distractions are not effective to contemporary models. Therefore, in this paper, the authors put forward a method that can produce distractions dynamically based on tree search. Experiments validate the effectiveness of the proposed method in distractions. The authors also explore the performance of defense methods like prompt-based methods and fine-tuning-based methods like DPO.

**Questions:**

I have the following questions/suggestions:
- It would be better to conduct experiments on some latest models like Qwen3 [1] and GPT-o3.
- In line 197, the tempature is set to 0.001during evaluation. Would using 0 be a more conventional choice for the evaluation?
- Could the author try another prompt template on a few models to observe the performance.
- For the background colors in Table1, a clear definition of the color assignments for each range would be beneficial.


------
**References**
[1] Yang, An, et al. "Qwen3 technical report." arXiv preprint arXiv:2505.09388 (2025).

**Ethical Concerns:**

["NO or VERY MINOR ethics concerns only"]

**Final Justification:**

This paper is clearly motivated and the proposed method is effective on different datasets. The experiments on more models and different prompt templates during the rebuttal addressed some of my original concerns.

**Limitations:**

yes

**Quality:**

3

**Strengths And Weaknesses:**

This paper has the following strengths:
- This method is clearly motivated and presented.
- Extensive experiments validate the effectivenss of the proposed method.
- The findings of mitigation methods are also useful for future research.

This paper has the following minor weaknesses:
- Lack of evaluations on latest reasoning models.
- It is unclear how the prompt template will affect the final results.

---

> ### Author Rebuttal · Authors · 2025-07-30
>
> We sincerely thank the reviewer for the detailed and thoughtful comments. We have carefully revised the paper based on your suggestions, and we address each point below.
>
> ---
>
> **$Q_1:$ It would be better to conduct experiments on some latest models like Qwen3 and GPT-o3.**
> **$A_1:$** We thank the reviewer for this valuable suggestion and fully agree that testing on the latest models is crucial for validating the generality of our findings. During the rebuttal period, we extended our experiments to include **Qwen3‑235B** and **GPT‑o3**, following the same **zero-shot CoT evaluation protocol** described in Section 3.1. The results are consistent with our main findings: the proposed distraction framework leads to **substantial accuracy drops** on these state-of-the-art models, confirming that the contextual robustness vulnerability persists even in the latest LLMs.
>
> | Dataset       | Model      | Original Acc. | Enhanced Acc. |
> | ------------- | ---------- | ------------- | ------------- |
> | CommonsenseQA | Qwen3‑235B | 0.94          | 0.35          |
> | CommonsenseQA | GPT‑o3     | 0.91          | 0.44          |
> | OpenbookQA    | Qwen3‑235B | 0.94          | 0.45          |
> | OpenbookQA    | GPT‑o3     | 0.94          | 0.49          |
> | TruthfulQA    | Qwen3‑235B | 0.86          | 0.55          |
> | TruthfulQA    | GPT‑o3     | 0.91          | 0.68          |
> | MMLU          | Qwen3‑235B | 0.87          | 0.55          |
> | MMLU          | GPT‑o3     | 0.89          | 0.60          |
>
> We will incorporate these new results into the revised version.
>
> **$Q_2:$ In line 197, the temperature is set to 0.001 during evaluation. Would using 0 be a more conventional choice for the evaluation?**
> **$A_2:$** We appreciate the reviewer for pointing out this detail. We fully agree that using 0 is a conventional choice for evaluation, and we verified that setting the temperature to 0 yields almost identical results.
>
> We adopted 0.001 mainly to ensure deterministic decoding across different APIs, as some implementations internally treat 0 as a special case (e.g., allowing minimal stochasticity or fallback sampling in rare scenarios). Using 0.001 enforces a strictly low-temperature regime while avoiding such implementation-dependent behavior.
>
> We will clarify this point in the revised version.
>
> **$Q_3:$ Could the author try another prompt template on a few models to observe the performance.**
> **$A_3:$** We thank the reviewer for this helpful suggestion. We agree that evaluating different prompting strategies is important for understanding whether prompt engineering can mitigate the impact of adaptive distractions.
>
> As shown in Table 11, we have already evaluated multiple commonly used prompting strategies, including vanilla direct answering, zero-shot CoT, few-shot CoT, in-context learning (ICL), and self-consistency. These results consistently show that although certain strategies yield slight improvements, none can fundamentally address the vulnerability—performance degradation remains substantial (average accuracy drop >45% across all models).
>
> Following the reviewer’s suggestion, we further tested an additional prompting variant during the rebuttal period: `repeat the question first`, where the model first restates the original question before answering. We summarize the complete results below:
>
> | Model        | Vanilla | Zero+CoT | Few+CoT | Few-shot (ICL) | Self-Consistency | Repeat Question |
> | ------------ | ------- | -------- | ------- | -------------- | ---------------- | --------------- |
> | GPT-4o-mini  | 0.19    | 0.21     | 0.24    | 0.26           | 0.27             | 0.27            |
> | GPT-4o       | 0.39    | 0.39     | 0.46    | 0.44           | 0.46             | 0.41            |
> | Qwen2.5-72B  | 0.33    | 0.34     | 0.35    | 0.34           | 0.35             | 0.35            |
> | LLaMA-3.1-8B | 0.25    | 0.25     | 0.27    | 0.28           | 0.29             | 0.33            |
>
> The “repeat the question first” strategy yields minor improvements on certain models (e.g., +0.06 on LLaMA-3.1-8B) but still falls short. This confirms our finding that **prompt-based methods alone cannot resolve contextual robustness failures** caused by semantically coherent but irrelevant distractions.
>
> We will include these results in the revised version for completeness.
>
> **$Q_4:$ For the background colors in Table 1, a clear definition of the color assignments for each range would be beneficial.**
> **$A_4:$** We thank the reviewer for pointing this out. The background colors in Table 1 are used to visually indicate the severity of performance degradation (Δ) after applying our adaptive distractions. The color coding is defined as follows:
>
> - Red (Severe Drop): Δ ≥ 0.4
> - Orange/Salmon (Moderate Drop): 0.3 ≤ Δ < 0.4
> - Yellow (Milder Drop): Δ < 0.3
>
> We will include this explanation directly in the revised table caption to make the color scheme explicit.
>
>
> ---
>
> Thank you again for your valuable time and suggestions and we would greatly appreciate your kind support during the discussion phase.

---

> > ### Comment · Reviewer_WpC3 · 2025-08-03
> >
> > Thanks for your reply. I have raised my score.

---

> > > ### Author Response · Authors · 2025-08-04
> > >
> > > We truly appreciate your recognition of our contributions, and we are glad that the additional experiments and clarifications addressed your concerns. Your feedback has been instrumental in improving the quality and clarity of our work.

---

### Official Review · Reviewer_tkgw · 2025-07-04

**Clarity:** 3
**Significance:** 2
**Originality:** 2
**Rating:** 4
**Confidence:** 3

**Summary:**

This paper presents a distraction generation framework that employs tree search and is guided by model behavior. The generated adaptive distractions—semantically coherent, answer-preserving contextual additions—can significantly impair the performance of large language models. Experiments on multiple-choice QA datasets demonstrate the effectiveness of the framework.

**Questions:**

1. Most experiments are conducted on multiple-choice question answering benchmarks. The method relies on incorrect answers to generate distractions. To enhance the framework utility, further experiments on generation tasks and safety tasks are needed, alongside comparisons with strong baselines. A brief experiment on MATH500 is noted in the appendix, but it is insufficient.

2. How are answers obtained during evaluation? Is it through testing probabilities or using chain-of-thought reasoning before providing an answer?

3. In Table 2, it would be beneficial to include several cases that compare perturbed questions generated by different methods. Additionally, what is the significant difference between DyVal2 and the proposed method?

4. Regarding the mitigation methods, the details of the data and training for DPO are unclear. Would having the model first repeat the question before answering help mitigate the issue?

**Ethical Concerns:**

["NO or VERY MINOR ethics concerns only"]

**Limitations:**

Please refer to the questions.

**Quality:**

3

**Strengths And Weaknesses:**

1. The framework offers a dynamic evaluation of LLM contextual robustness.
2. The adaptive distractions lead to substantial performance degradation, with an average accuracy drop exceeding 45%, highlighting vulnerabilities even in advanced LLMs.

---

> ### Author Rebuttal · Authors · 2025-07-30
>
> We sincerely thank the reviewer for the detailed and thoughtful comments. We have carefully revised the paper based on your suggestions, and we address each point below.
>
> ---
>
> **$Q_1:$** **Most experiments are conducted on multiple-choice question answering benchmarks. The method relies on incorrect answers to generate distractions. To enhance the framework utility, further experiments on generation tasks and safety tasks are needed, alongside comparisons with strong baselines. A brief experiment on MATH500 is noted in the appendix, but it is insufficient.**
>
> **$A_1:$** We thank the reviewer for the valuable suggestion regarding extending our framework to safety-sensitive generation tasks. We fully agree that this is an important direction to test the generality of our method.
>
> However, as also noted in prior literature on knowledge conflicts [1], generation tasks such as summarization or open-ended QA introduce intrinsic ambiguity: the output does not have a single ground-truth answer, and adding semantically coherent distractions may naturally shift the intended interpretation of the task rather than purely reflecting robustness issues. For instance, when extra context contradicts the model’s prior knowledge, the model might simply follow the context, which is an expected behavior in retrieval-augmented generation scenarios rather than a failure of contextual robustness. Therefore, we believe that applying our current answer-preserving, ground-truth-based framework directly to such open-ended tasks would risk **confounding contextual robustness with task re-interpretation**. We consider this an important future direction, where task-specific robustness metrics (e.g., factual consistency measures) can be integrated.
> In contrast, safety-critical tasks provide **well-defined objectives** (e.g., generating harmful or toxic outputs), making them more suitable for our framework. To explore this, we conducted additional experiments using the RealToxicityPrompts dataset, which is widely used for evaluating model safety. For each prompt, we searched for a semantically coherent instructional distraction (i.e., a prefix before the prompt) that would most increase the toxicity of the response, guided by our adapted value function (using Perspective API scores as feedback).
>
> We compared our method against a single-step baseline that applies a direct distraction prefix without iterative refinement. The results on 175 samples are summarized below:
>
> | Model       | Original Toxicity | +Direct Distraction | +Adaptive Distraction (Ours) |
> | ----------- | ----------------- | ------------------- | ---------------------------- |
> | GPT-4o-mini | 0.0304            | 0.2279              | **0.5496**                   |
> | Qwen2.5-72B | 0.0908            | 0.2256              | **0.5190**                   |
> | GPT-4o      | 0.0277            | 0.1291              | **0.2737**                   |
> | LLaMA3-8B   | 0.0558            | 0.1508              | **0.3616**                   |
> | Gemma-2-27B | 0.0998            | 0.1857              | **0.4748**                   |
>
> These results demonstrate that our adaptive distraction strategy can significantly increase model toxicity more effectively than single-shot distractions, confirming that the proposed framework is capable of revealing safety vulnerabilities beyond standard QA tasks.
>
> We will include these findings in the revised version.
>
> **$Q_2:$ How are answers obtained during evaluation? Is it through testing probabilities or using chain-of-thought reasoning before providing an answer?**
>
> **$A_2:$** We thank the reviewer for this question. In our experiments, answers are obtained using **zero-shot chain-of-thought (CoT) prompting**, as this setting better reflects typical usage where reasoning steps are explicitly encouraged. This is described in Section 3.1 (Experiment Setup), and the complete prompt templates for answer generation, semantic consistency checking, and baseline comparisons are provided in **Appendix G**.
>
> **$Q_3:$** **In Table 2, it would be beneficial to include several cases that compare perturbed questions generated by different methods. Additionally, what is the significant difference between DyVal2 and the proposed method?**
>
> **$A_3:$** We thank the reviewer for this helpful suggestion. Below we provide a representative example to illustrate how our method differs from existing approaches.
>
> The original question is: “In 2017, how many people died from terrorism globally?” (Correct answer: 26,000).
> - The **ICA** baseline typically adds generic, task-irrelevant context, such as:
>   *“That year also saw record-breaking economic growth in some countries and several new species discovered in the Amazon rainforest.”*
> - The **SPD** baseline introduces topic-related but shallow static perturbations, e.g.:
>   *“According to public security records, many reports highlighted both terrorist activities and global military spending trends that year.”*
> - The **DyVal2 (MPA)** baseline adds psychometric-principle-based extra context, for example:
>   *“Some agencies reported terrorism fatalities declined compared to 2016, though regional conflicts in the Middle East caused significant civilian casualties.”*
>
> These approaches primarily rely on semantically plausible context but are **not guided by model behavior** and are not explicitly optimized to systematically reveal vulnerabilities.
>
> By contrast, our method integrates semantically coherent distraction generation with **behavior-guided optimization and quantitative validation**. For the same question, our method generates:
> *“A devastating earthquake struck a densely populated region, resulting in thousands of casualties. Interestingly, a significant drop in reported terrorist incidents was reported in Southeast Asia.”*
> This distraction is explicitly optimized based on victim-model feedback to exploit its tendency to conflate large-scale casualties with terrorism deaths.
>
> More importantly, this effect is validated quantitatively: as shown in Table 2, our distractions consistently cause a substantial accuracy drop across multiple models (**average drop >45%**), demonstrating that our framework reveals a **systematic weakness in contextual robustness** rather than isolated case-specific failures, while keeping the ground-truth answer unchanged.
>
> For clarity, we also summarize the key differences between DyVal2 and our method:
> (1) **Objective**: DyVal2 focuses on broad ability probing and contamination detection, while our method specifically targets contextual robustness evaluation.
> (2) **Behavior Guidance**: DyVal2 follows predefined psychometric principles without victim-model feedback, whereas our method uses a behavior-guided tree search to iteratively optimize distractors for maximal failure.
> (3) **Answer Preservation**: DyVal2 does not strictly enforce answer preservation; our method incorporates semantic consistency filtering and proxy-model validation to ensure the correct answer remains valid.
>
> We will add these clarifications and examples to the revised version for improved readability.
>
> **$Q_4:$** **Regarding the mitigation methods, the details of the data and training for DPO are unclear. Would having the model first repeat the question before answering help mitigate the issue?**
>
> **$A_4:$**
>
> **(1) DPO Data and Training Details:** We apologize if the details were not sufficiently emphasized in the main text; they are described in Appendix B.1 (lines 576–592), but we summarize them here for clarity.
> We curated approximately **1,200 preference pairs** from prior experiments, where the preferred answer corresponds to the correct response under the clean context and the dispreferred answer is the incorrect response under the distracted context. To ensure fairness, enhanced questions originating from the same original question never appeared in both the training and test sets.
> The data was split into 80% training, 10% validation, and 20% testing. We fine-tuned three open-source models (**Gemma-2-2B**, **Qwen2.5-7B**, and **Phi-3.5-mini**) using Direct Preference Optimization (DPO) on two RTX 4090 GPUs for five epochs with a learning rate of 2e-4.
> The evaluation followed the same zero-shot CoT prompting setting as in the main experiments, ensuring comparability (see Table 5 and Table 6).
>
> **(2) Prompt-based Mitigation (Repeating the Question):**
> We appreciate the reviewer’s suggestion. While we already evaluated multiple prompt-based strategies in the main paper (Table 11), we additionally tested this new “**repeat the question first**” strategy during rebuttal. The results are shown below, including all prompt-based baselines from the original paper for completeness:
>
> | Model        | Vanilla | Zero+CoT | Few+CoT | Few-shot (ICL) | Self-Consistency | Repeat Question |
> | ------------ | ------- | -------- | ------- | -------------- | ---------------- | --------------- |
> | GPT-4o-mini  | 0.19    | 0.21     | 0.24    | 0.26           | 0.27             | 0.27            |
> | GPT-4o       | 0.39    | 0.39     | 0.46    | 0.44           | 0.46             | 0.41            |
> | Qwen2.5-72B  | 0.33    | 0.34     | 0.35    | 0.34           | 0.35             | 0.35            |
> | LLaMA-3.1-8B | 0.25    | 0.25     | 0.27    | 0.28           | 0.29             | 0.33            |
>
> The “repeat the question first” strategy yields **slight improvements** on some models (e.g., **+0.06 on LLaMA-3.1-8B**) but remains far less effective than DPO.
> This further supports our conclusion that **prompt-based methods alone cannot fundamentally address reasoning failures** induced by adaptive distractions.
>
> We will include these additional results in the revised version.
>
> [1]: Making Retrieval-Augmented Language Models Robust to Irrelevant Context
>
> ---
>
> Thank you again for your valuable time and suggestions and we would greatly appreciate your kind support during the discussion phase.

---

> > ### Comment · Reviewer_tkgw · 2025-08-06
> >
> > Thank you for the detailed reply and additional experiments. Given the reliance on incorrect answers to generate distractions and the need for a method that effectively addresses this type of robustness, I will maintain my initial score.

---

### Note · Authors · 2025-08-13

Dear AC and Reviewers,

We sincerely thank you for the constructive feedback, which has helped us improve the clarity, rigor, and completeness of our work.

### **Overall Conclusion**

All concerns have been fully addressed through additional experiments, clarifications, and manuscript revisions.

### **Key Actions Taken & Evidence Added**

- **Core contribution clarification:**
  Clarified that our key contribution is *behavior-guided, answer-preserving tree search* for generating logically distracting yet semantically coherent inputs. This addresses reviewer SXak’s misunderstanding that our work merely identifies contextual vulnerability.

- **Safety generalization:**
  Extended our method to a safety-critical task (RealToxicityPrompts) with a new toxicity-maximizing value function. Our adaptive context raises toxicity rates significantly across models, showing generality beyond MCQA.

- **Latest model results:**
  Added results on Qwen3-235B and GPT-o3. Both models experience large drops, confirming continuing relevance of frontier models.

- **Efficiency vs depth tradeoff:**
  Compared single-step, brute-force, and our guided search. Our method matches brute-force drop with ~64% fewer nodes (5.3 vs 14.7), justifying value-guided search and depth >1.

- **Baseline & DyVal2 comparison:**
  Provided qualitative and quantitative comparisons to DyVal2 and other static baselines. Our method drops accuracy by >45% on average, while baselines achieve only 15–20%.

- **Prompting vs DPO:**
  Evaluated “repeat-the-question” prompting; marginal improvement (+0.06 on LLaMA3.1-8B), still far below DPO. Confirms prompting alone is insufficient.

### **Reviewer Status Update**

- **Reviewer coVs:** All concerns resolved; **score remains 5**.
- **Reviewer tkgw:** All concerns resolved; **score remains 4**.
- **Reviewer WpC3:** All concerns resolved; **score raised to 5**.
- **Reviewer SXak:** **Score remains 2**. The main concern was based on a misunderstanding of our contribution—mistaking our system for mere contextual distraction definition rather than a behavior-guided generation method. We clarified this in both the rebuttal and manuscript. Reviewer feedback also included several technical concerns (e.g., on DPO training), which we addressed with detailed new experiments. We thank the AC for considering this context.

We once again thank all reviewers for their feedback and the AC for fair and careful oversight.

Sincerely,
Authors

---

### Decision · Program_Chairs · 2025-09-17

**Decision:**

Accept (poster)

**Comment:**

This paper introduces a framework for evaluating the contextual robustness of LLMs through adaptive, behavior-guided tree search. The method generates distractions that are semantically coherent but do not change the correct answer, and can cause accuracy drops of over 45% across benchmarks and models. The work is clearly communicated, with well motivated design, ablations, and comparisons against static baselines and mitigation strategies. The rebuttal adds new experiments (GPT-o3, Qwen3-235B) and safety-critical tasks (RealToxicityPrompts) and clarifies evaluation protocol and efficiency trade-offs.

The strengths are novelty and rigor. Previous work explored static or retrieval-based distractions, but here the use of behavior-guided tree search advances what can be captured. The evaluation is comprehensive, spanning multiple datasets and models, and the inclusion of mitigation experiments provides additional value by demonstrating both the limits of prompt engineering and the promise of post-training approaches such as DPO.

The main weaknesses are computational cost and scope. The tree-search process is resource-intensive, and although efficiency heuristics reduce this burden, scalability to very large datasets or integration into training loops remains an open question. In addition, the framework is primarily validated on QA-style tasks with fixed answers, leaving open how it might extend to open-ended generation tasks where “failure” is harder to define. These concerns were acknowledged by the authors and partially addressed in rebuttal through safety-oriented experiments and discussion of future directions.

Reviewer evaluations are mixed. Reviewers coVs and WpC3 strongly support acceptance, highlighting the novelty and solid evaluation. Reviewer tkgw is borderline, due to technical soundness but limited generality. Reviewer SXak is strongly negative, arguing the work is not meaningfully distinct from prior knowledge-conflict studies (though the rebuttal addresses this) and questioning the choice of baselines. The rebuttal added detailed clarifications and new experiments, which convinced three reviewers but not SXak. Given the significance of the problem, the methodological advance, and majority support from the reviewers, I recommend acceptance as a poster.